# 'It takes two to tango': Bridging the gap between country need and vaccine product innovation

Rachel A. Archer[1]*, Ritika Kapoor[2], Wanrudee Isaranuwatchai[1,3], Yot Teerawattananon[1,2], Birgitte Giersing[4], Siobhan Botwright[4], Jos Luttjeboer[5,6], Raymond C. W. Hutubessy[4]

1 Health Intervention and Technology Assessment Program, Ministry of Public Health, Nonthaburi, Thailand, 2 Saw Swee Hock School of Public Health, National University of Singapore, Singapore, Singapore, 3 Institute of Health Policy, Management and Evaluation, University of Toronto, Toronto, Canada, 4 Immunization, Vaccines and Biologicals Department, World Health Organization, Geneva, Switzerland, 5 Department of Medical Microbiology, University Medical Center Groningen, University of Groningen, Groningen, The Netherlands, 6 Asc Academics, Groningen, The Netherlands

* Rachel.a@hitap.net

**Data Availability Statement:** All relevant data are within the paper and its Supporting Information files.

## Abstract

### Background

Despite a growing global commitment to universal health coverage, considerable vaccine coverage and uptake gaps persist in resource-constrained settings. One way of addressing the gaps is by ensuring product innovation is relevant and responsive to the needs of these contexts. Total Systems Effectiveness (TSE) framework has been developed to characterize preferred vaccine attributes from the perspective of country decision-makers to inform research and development (R&D) of products. A proof of concept pilot study took place in Thailand in 2018 to examine the feasibility and usefulness of the TSE approach using a rotavirus hypothetical test-case.

### Methods

The excel-based model used multiple-criteria decision analysis (MCDA) to compare and evaluate five hypothetical rotavirus vaccine products. The model was populated with local data and products were ranked against decision criteria identified by Thai stakeholders. A one-way sensitivity analysis was performed to identify criteria that influenced vaccine ranking. Self-assessment forms were distributed to R&D stakeholders on the usability of the approach and were subsequently analysed.

### Results

The model identified significant parameters that impacted on MCDA rankings. Self-assessment forms revealed that TSE was perceived as being able to encourage closer collaboration between country decision makers and vaccine developers.

**Funding:** This pilot study was funded by the Bill and Melinda Gates Foundation. The Health Intervention and Technology Assessment Program (HITAP) is funded by the Thailand Research Fund under a grant for Senior Research Scholar (RTA5980011). HITAP's International Unit is supported by the International Decision Support Initiative (iDSI) to provide technical assistance on health intervention and technology assessment to governments in low- and middle-income countries. iDSI is funded by the Bill & Melinda Gates Foundation [OPP1202541], the United Kingdom's Department for International Development, and the Rockefeller Foundation. The funders had no role in study design, data collection and analysis, decision to publish, or preparation of the manuscript. HITAP's International Unit collaborates with The Access and Delivery Partnership.

**Competing interests:** The authors have declared that no competing interests exist.

**Abbreviations:** CAPACITI, Capacity-led Assessment for Prioritisation on Immunisation; HITAP, Health Intervention and Technology Assessment Program; HTA, Health Technology Assessment; LMICs, Low-And Middle-Income Countries; MCDA, Multiple-Criteria Decision Analysis; NITAG, National Immunization Technical Advisory Group; NUS, National University of Singapore; R&D, Research and Development; TPP, Target Product Profile; TSE, Total Systems Effectiveness; UHC, Universal Health Coverage; WHO, World Health Organization.

## Conclusions

The pilot study demonstrates that it is feasible to use an MCDA approach to elicit stakeholder preferences and determine influential parameters to help identify the preferred product characteristics for R&D from the perspective of country decision-makers. It found that TSE can help steer manufacturers to develop products that are better aligned with country need. Findings will guide further development of the TSE concept.

## Introduction

All United Nations Member States have agreed to accelerate progress towards universal health coverage (UHC) by 2030 [1]. Today, at least half of the world's population do not have full access to essential health services. Each year, 20 million infants do not receive their full course of recommended vaccines, resulting in approximately 1.5 million deaths from vaccine preventable diseases. Over 50% of these infants live in six countries with the weakest health systems infrastructure [2]. In terms of inequitable health coverage, it is estimated that 100 million people live in extreme poverty due to out of pocket expenditure on health [3]. Effective global immunisation is essential to attaining the UHC goal, however to achieve this, vaccines must be designed suitable for use in resource constrained settings. To have the greatest public health impact, the product attributes of vaccines that are intended for deployment in low-and middle-income countries (LMICs) must be informed by the needs of LMIC vaccine delivery systems.

LMICs are confronted with unique challenges when implementing successful immunisation programmes [4–6]. Persisting difficulties include but are not limited to: logistical complexity, barriers and gaps in delivery systems, constraints in human resource and challenges in prioritisation of vaccines over other competing interventions. Issues of ensuring timely delivery and appropriate storage conditions to preserve vaccine potency and safety prevail. In addition, the emergence and introduction of new vaccines, improved vaccine products and novel delivery technologies requires that policy and decision makers have the information and tools to assess and select the products that will be of greater impact for their programmes, after product licensure as well as during product development, to ensure that product characteristics are aligned with country need.

The above challenges faced by each country may vary significantly based on their geographical, healthcare, and political constraints [7]. This impacts the priorities and goals for vaccine programs of every country and differentiated vaccine product characteristics may be desirable to overcome these constraints [8]. Despite wide acknowledgment of these practicalities, there is no clear methodological approach of identifying the vaccine product attributes that would address these context specific barriers, and communicating these country priorities to inform vaccine R&D by manufacturers.

R&D describes the discovery and technical advancement of a new or existing technology [9]. The development of novel vaccine products and innovative delivery mechanisms is vital to address to the current programmatic challenges and combat the threats of emerging and existing infectious diseases. Vaccine R&D is not an easy endeavor; vaccines are complex and costly biological preparations. The development and introduction of one new vaccine product requires immense amounts of money, time, and resources and there is no guarantee of a return on this investment [10, 11]. Aligning the vaccine design and development initiatives with the government vaccine product preferences will likely help to incentivise and accelerate the

development of vaccines that are optimal for local contexts, but these vaccines may also have broader potential for uptake and impact at the regional or global level. Designing new vaccine products that better anticipate market demand and challenges and are more responsive to government needs, will decrease the risk in investment, and increase the potential for return.

Early stage Health Technology Assessment (HTA) is the evaluation of new technologies in development to determine their potential value and provides an opportunity to facilitate dialogue between government and market actors. Performing such an evaluation with inputs from key stakeholders including decision makers, immunisation programme managers, clinical experts, and manufacturers can help to identify preferred vaccine attributes, thereby informing R&D about the key needs of innovation and accelerating product uptake [12].

The World Health Organization (WHO) and partners are developing the concept of Total Systems Effectiveness (TSE), which aims to strengthen structured and transparent vaccine product selection processes at the country level and to build a platform for stakeholders in LMIC immunisation programmes to shape vaccine supply and R&D. TSE has since been renamed Capacity-led Assessment for Prioritisation on Immunisation (CAPACITI) [13]. The aim of the TSE approach is to foster a stronger collaboration between industry and country decision makers to develop products that are more closely aligned with the needs and context of LMICs.

The WHO TSE framework is designed to support the evaluation of different vaccine products for product selection at the country level and is based on multi-criteria decision fanalysis (MCDA), in order to consider trade-offs between different product characteristics. Whilst WHO envisages utilising the tools within its TSE framework to assess vaccine products, the framework approach itself is based on a generic concept that could be adapted to other applications, e.g. drugs and medical devices. Whilst the literature reports an increase in application of MCDA for healthcare evaluations related to a range of decisions including reimbursement, prescription and resource allocation decisions, there is limited documented use of MCDA to identify priorities for R&D based on LMIC need [14–16].

To test the assertion that the TSE approach could inform R&D of vaccine products for LMICs, WHO, the Health Intervention and Technology Assessment Program (HITAP) and National University of Singapore (NUS), piloted TSE in Thailand between April and August 2018. The objective of this article is to explore the application of TSE approach to identify the preferred product characteristics of a vaccine product that is preferable from the perspective of decision makers in Thailand, and to assess how this can be used to inform the country's manufacturers for the R&D process. Firstly, this study examines whether the TSE approach can be used to identify important characteristics for a vaccine product to have market advantage over existing products. Secondly, this study also assesses the perceived usefulness of the TSE approach among stakeholders in three Southeast Asian countries, namely Indonesia, Thailand, and Vietnam. The pilot study used five example Rotavirus vaccine products as a hypothetical test case, because of the variable product characteristics of both existing, licensed and pipeline rotavirus vaccines.

## Methods

Three workshops were held during the pilot to better understand criteria of importance in national decision-making and the perspective of different stakeholders towards TSE. An initial workshop in May 2018 was convened to obtain preliminary feedback from stakeholders involved in vaccine policy decision making and research in Thailand. In addition to WHO, HITAP and NUS, 15 participants were in attendance and included representatives from: Sub-committee of the National List of Essential Medicines, The National Immunization Technical

Advisory Group (NITAG), academia, pharmaceutical industry and other ministerial departments. During the workshop, stakeholders were asked to complete an open-ended survey questionnaire to nominate top criteria important for vaccine product selection. The survey template was used previously by the WHO in a stakeholder meeting convened in Indonesia. The survey instrument was subsequently fine-tuned and modified for this study context. The survey instrument is available in S1 File. Responses were collected from 15 respondents and the top five criteria were selected as the decision-making criteria for the pilot study in Thailand.

For the purpose of the study, an excel-based model for rotavirus product selection, based on the TSE approach, was modified for the Thai context. The TSE rotavirus product selection model included a generic set of decision criteria, with defined indicators, and enabled comparison between five hypothetical rotavirus vaccine products (RVV1 to RRV5, product characteristics detailed in S1 Table). The final list of decision criteria for the Thai model was identified by stakeholders in Thailand. The final five criteria and associated outcomes for the Thai model are as follows: 1) health outcomes (i.e. cases averted, hospitalisation cases averted, and deaths averted due to the vaccination); 2) cost estimates (total programme costs, healthcare costs, and a five-year budget impact); 3) safety data (intussusception cases); 4) budget impact; and 5) cost effectiveness. Since budget impact and cost-effectiveness had not been included in the generic TSE rotavirus model, they were added for the purpose of the exercise.

The excel-based model utilized methodology that has been used in established models such as UNIVAC and the Vaccine-Technology Impact Assessment (V-TIA) tool to calculate the outcomes defined for each criterion, using country and product specific data inputs [17]. Locally relevant parameter inputs were generated from government reports, published literature and expert opinions. They also included local inputs on socio-economic status, coverage of vaccine programs, vaccine efficacy, and schedule, costs for storage, training and administration, and other epidemiological data on birth cohort, disease burden and resource use (S2 Table).

Following MCDA methodology, the outcomes for each criterion were transformed into a score based on a common absolute scale ranging from 0–100, to calculate an aggregate score for each option, resulting in a final ranking of each options according to performance across all criteria [18]. A vaccine with a higher total score was considered to be the better choice. Equal weights were used for all criteria, as the research team were unable to collect information on stakeholder preferences for weighting. A full description of the model and how the model outcomes were generated is available elsewhere [17].

A one-way sensitivity analysis was conducted on the vaccine characteristics, to identify the thresholds of each characteristic that could modify the overall ranking of products. It is hypothesised that this could form the basis for identifying minimum and aspirational bounds for target product profiles that meet country needs. We chose the vaccine product which is currently being ranked the second in the MCDA analysis, and via the sensitivity analysis, investigated if by varying the vaccine characteristics, it could become the top ranked vaccine. The sensitivity analysis was restricted to base and best case values for vaccine characteristic for identifying the variable changes which could make the second best product become the best ranked product. The base case was determined by the input values that reflected the most likely scenario and the input values for the best case represented the most optimistic scenario. The details on the inputs for the vaccine related variables for the base case and the best case can be found in the S3 Table.

To identify the usefulness of such an approach from the R&D perspective, a separate workshop was held with stakeholders working in R&D in LMICs from both public and private sector organisations in Indonesia, Thailand, and Vietnam. The results from the rotavirus product

MCDA evaluation were disseminated to illustrate the principle that–with information on country priorities for vaccine products–manufacturers could model a pipeline product against those of competitors to identify important attributes and thresholds given uncertainty. Key challenges and priorities in the countries were presented, and discussions were encouraged to deliberate the relevance and usability of TSE. Qualitative data was also collected from 17 self-assessment forms on the applicability and usefulness of this approach to inform R&D. The self-assessment form included a mixture of closed and open-ended questions to gather participant opinions of TSE and is available in S2 File. The form was developed for the purpose of this meeting and participation was completely voluntary. Participants were made aware that the forms were anonymized and that responses were to be kept strictly confidential and reflect personal opinions rather than the positions of their employers. Thematic analysis was deployed to analyse main patterns in the responses within the questionnaires. The forms were read repeatedly to enhance the overall understanding, then coded and classified. Then codes were grouped into emerging themes. A third workshop was convened to present the pilot study results back to the stakeholders who attended the initial meeting.

## Results

### Criteria selected by decision makers in Thailand

The top 5 decision-making criteria identified by stakeholders for choosing between rotavirus vaccine products were as follows: health impact, safety, budget impact, cost-effectiveness and delivery costs. The different outcome measures which represented the criteria are illustrated in Table 1. The scoring of the vaccine products reported RVV-3 to be the most preferred vaccine product, followed by RVV-2 as the second rank vaccine product followed by RVV-4, RVV-5 and RVV-1 as the rank 3rd, 4th, and 5th ranked vaccine product respectively (S4 Table).

### Vaccine characteristics influencing the vaccine performance on decision criteria

Multiple vaccine characteristics were then identified which could impact the scores of the vaccine products on the decision criteria nominated by stakeholders. One vaccine characteristic could influence the vaccine performance in one or more decision criteria since decision criteria are not mutually exclusive/interdependent. The different vaccine characteristics included the volume, number of doses and doses per vial, cooling method, commodity cost, efficacy, duration of protection, relative risk of intussusception, and vaccine schedule, see Table 2. The non-vaccine parameters included disease epidemiological data (e.g. disease burden, severity of the disease, associated mortality and numbers of outpatient and hospitalisation), system cost data (training costs, salaries of health care workers, cost of hospitalisations), and other parameters such as coverage and socio-economic status.

**Table 1. Top 5 decision making criteria for Thailand stakeholders for vaccine policy making.**

| Decision Criteria | Description of related outcome measure |
| --- | --- |
| Safety | Intussusception hospitalisations due to vaccine |
| Health Impact | Rotavirus cases averted due to vaccination |
| Budget Impact | Overall 5-year budget impact including the cost of program and healthcare cost* |
| Delivery cost | Transport and storage costs for vaccines |
| Cost-effectiveness | Incremental costs per DALY |

* The overall budget impact of the vaccine includes the cost of the immunization program and the also the healthcare resources spent on the rotavirus cases and the intussusception cases the population experiences.

Table 2. A breakdown of the influential vaccine characteristics.

| Decision criteria | Vaccine characteristics influencing criteria | Other parameters influencing criteria |
|---|---|---|
| Safety | Relative risk of intussusception, Number of doses, Vaccine schedule | |
| Health impact | Vaccine efficacy, Number of doses, Duration of protection, Vaccine schedule | Socio-economic status, Coverage |
| Budget impact | Commodity cost, Volume of vaccine, Method of cooling, Number of doses per vial, Vaccine efficacy, Number of doses, Duration of protection, Vaccine schedule, Relative risk of intussusception | Training costs, Trends in size of birth cohorts, Trends in coverage, Salary additional health care workers |
| Delivery cost | Vaccine volume | Electricity price, Petrol price, Number of deliveries, Distance between the level in the supply chain |
| Cost-effectiveness | Commodity cost, Volume of vaccine, Method of cooling Number of doses per vial, Vaccine efficacy, Number of doses, Duration of protection, Vaccine schedule, Relative risk of intussusception | Rotavirus incidence, Proportion severe, Rotavirus mortality, Coverage of existing immunisation schedules, Socio-economic status, Number of deliveries, Training costs & salary of healthcare workers, Distribution of inpatient visits over different levels of health care, Distribution of outpatient visits over different levels of health care, Length of hospitalization, Intussusception case fatality rate, Cost of cooling |

## Sensitivity analysis to identify significant vaccine characteristics

Varying certain vaccine characteristics independently did change the rank order of RVV-2 from second to first position in the rotavirus product selection model. Vaccine characteristics that changed the rank order of RVV-2 are as follows: relative risk of intussusception, number of doses, vaccine efficacy, duration of protection and commodity cost. The best-case scenario of each of these significant parameters could independently make RVV-2 the top ranking vaccine (Table 3). Details on the impact of the best case scenarios used for the one-way sensitivity analysis on the overall vaccine scores can be found in the S5 Table.

## Usefulness of TSE approach to inform vaccine R&D: R&D stakeholder perspective

Several themes transpired from the participant's self-assessment including: potential to strengthen product selection process, limited local capacity, need for technical assistance,

Table 3. Sensitivity analysis to identify significant vaccine characteristics for making RVV-2 the highest ranked vaccine.

| Vaccine Characteristics | Inputs for RVV-2 | | Old vaccine Rank | New Vaccine Rank | Related decision criteria |
|---|---|---|---|---|---|
| | Base-case* | Best-case** | | | |
| Relative risk of intussusception | 1.24 | 1.0 | 2nd | 1st | Safety, Budget Impact, Cost-effectiveness |
| Number of doses | 2 | 1 | 2nd | 1st | Safety, Health Impact, Budget Impact, Cost-effectiveness |
| Vaccine Schedule | DTP-1 (6 weeks after birth) | OPV-1 (1 week after birth) | 2nd | 2nd | Safety, Health Impact, Budget Impact, Cost-effectiveness |
| Vaccine efficacy | 50% | 100% | 2nd | 1st | Health Impact, Budget Impact, Cost-effectiveness |
| Duration of Protection (weeks) | 52 | 156 | 2nd | 1st | Health Impact, Budget Impact, Cost-effectiveness |
| Commodity cost (US$) | 2.2 | 1.1 | 2nd | 1st | Budget Impact, Cost-effectiveness |
| Volume of the vaccine (m$^3$) | 17.6 | 8.8 | 2nd | 2nd | Budget Impact, Delivery costs, Cost-effectiveness |

* The most likely input value

** The best or aspirational input value

**Table 4. Profile of respondents from the self-assessment forms.**

|  | Number |
|---|---|
| **TYPE OF ORGANISATION** |  |
| **GOVERNMENT** | 8 |
| **PRIVATE SECTOR** | 5 |
| **INTERNATIONAL ORGANISATION** | 3 |
| **NOT SPECIFIED** | 1 |
| **COUNTRY OF WORK (THAILAND/OTHER/BOTH)** |  |
| **THAILAND** | 11 |
| **OTHER** | 3 |
| **BOTH** | 2 |
| **NOT SPECIFIED** | 1 |

concerns of applicability. The profile of respondents is documented in Table 4 and the detailed results of the thematic analysis can be found in S6 Table.

The majority of stakeholders stated that characteristics for new vaccine products are being decided unsystematically and with limited visibility of country priorities, and unclear rationale for selection, especially for markets outside of their country. There was broad agreement that TSE could have a pivotal role in providing a platform for discussion between public health and R&D stakeholders to create alignment regarding preferred characteristics of products. As such, it was felt that TSE could potentially be an approach to develop an R&D strategy for countries to drive innovation and increase the likelihood of uptake beyond national markets, by considering attributes that would facilitate recommendations and uptake in other market segments.

One limitation raised was that the TSE approach will only collect information from countries that are implementing the TSE framework for product evaluation and decision-making, which requires developing the technical capacity for TSE implementation. For TSE to be applicable in many settings, there is a need to strengthen the country's technical capacity for evidence-informed priority setting. Shifts in decision making priorities from the time of initiation of product development to the phase of market translation of the finished product was another concern on the usability of TSE. The R&D stakeholders maintained that it is of great importance that TSE is regularly revisited and revised to reflect changing country specific priorities.

## Discussion

This study was based on a proof-of-concept exercise to find out whether an MCDA approach to product selection could analyse trade-offs in characteristics of pipeline products if stakeholder perspectives are known. The findings illustrated that an understanding of national decision-maker priorities for vaccine policy or selection decisions can help to characterize preferred product attributes. Varying either of the following vaccines characteristic; relative risk of intussusception, number of doses, vaccine efficacy, and duration of protection and commodity cost, independently by giving it the best-case value (based on the most optimistic scenario) could propel the second-ranked vaccine into first rank in the rotavirus product selection model. This demonstrates the importance of these characteristics to the Thai stakeholders consulted.

Understanding product preferences from the perspective of country decision makers provides critical input into the Target Product Profile (TPP) for vaccine products. TPPs describe the minimally acceptable and aspirational ranges of product features to guide product

development of vaccines, and establish go/no-go criteria for prioritisation and investment by developers. As such, the TSE approach could allow manufacturers to understand country needs, which could be combined with technical considerations such as likelihood of clinical success and manufacturing feasibility to develop more robust TPPs and roadmaps for products with higher chance of country uptake that will ultimately lead to greater public health impact [19].

This study demonstrated the application of MCDA as systematic approach for evaluating the relative value proposition, or trade off, of existing and potential products based on country priorities. It is no longer considered that a single criterion, such as cost-effectiveness or disease burden, is sufficient to rank vaccines in the pipeline; a multitude of factors come into play [20, 21]. MCDA methodology allows for input from a wide range of stakeholders with an array of priorities to support effective vaccine development decisions [21]. Application of criteria weights which represent the relative importance of the decision criteria further imparts flexibility to adapt the approach to different diseases and country setups [22]. The study used equal weighting and did not include varied weights to represent the importance of the decision criteria for the stakeholder group nor perform sensitivity analysis to the equal weights assigned to the criteria due to lack of relevant data collection. However, the authors feel that altering the distribution of the weights in such evaluations would be of significant importance to better understand the robustness of the characteristics. Furthermore, the outcomes associated with each criterion were already pre-determined in the model. Allowing stakeholders to define their own outcomes may lead to a better understanding of expectations for pipeline products from a country perspective.

Special attention needs to be paid when determining the distribution of values identified to test best- and worst-case scenarios. A small range may inflate the minimal advantage offered by a vaccine product over the other products, making it the best scoring vaccine. Similarly, for decision criteria where even small changes are important like deaths avoided, a wide scoring range, may deflate the benefits on the scoring scales. An important requirement of MCDA is to engage multiple stakeholders to identify decision criteria and ensure the results reflect national priorities [23]. TSE can only be applicable if it is rooted in a structured and transparent process with an explicit rationale behind decisions.

TSE takes an early HTA approach to systematically evaluate the value of vaccine products in the pipeline. Utilising HTA during R&D stage, when the major product design decisions are made, can be greatly beneficial to manufacturers. Traditional HTA is applied to existing products on the market. Products that fail this late-stage HTA waste millions of dollars in lost investment, and risk increase market entry costs for future products. Performing early stage HTA, particularly in the form of the TSE framework, provides an opportunity to inform manufacturer TPPs which can mitigate this uncertainty [24], de-risking R&D investment and accelerating product utilisations.

This study uses one-way sensitivity analysis to emphasize how single vaccine characteristics may be significant in favouring one vaccine product over the other. A one-way sensitivity analysis is in line with the standard approach in the literature for early health technology assessment (HTA) [24]. This approach used in this study is named headroom method is a way of estimating the maximum reimbursable price of the new device over a comparator to determine a value-based price ceiling. While multivariate and probabilistic sensitivity analysis are widely accepted in traditional HTA, the authors have not seen any guidelines of how it can be applied to early HTA. This is an area for future research.

The TSE approach is not without its limitations. In order for TSE to inform product selection, and product optimisation, the framework needs to be embedded within, and must complement the country's existing vaccine evaluation process. Our study used a quantitative

model, whereas the TSE framework has been adapted to function in environments with varying data quality and analytical resources.

Also, the current study findings are limited to the local conditions and decision-making mechanisms of Thailand and may not be generalisable to other developing countries in the world. Stakeholders consulted were a small sample of individuals across the Southeast Asian region and predominantly from the public sector. Findings will not be representative of the experiences of all those involved in vaccine innovation and product selection decisions in the region, and further study needs to be undertaken to consider whether information from using the TSE framework in data poor environments yields similar interest from R&D stakeholders.

## Conclusion

This studied verified that a critical communication gap between country decision-makers and vaccine developers exists in the South-East Asia region, and findings illustrate that the TSE approach may be used to identify the priority vaccine attributes of country stakeholders through MCDA and early HTA. The authors believe that shifting to a 'needs-driven' R&D paradigm can move mountains in addressing preventative health needs and accelerating global health progress towards Universal Health Coverage. Successful implementation of this approach would be a win for all the stakeholders–government accessing products tailored to country needs, manufacturers enjoying a lower risk in R&D and patients attaining a faster access to good quality health benefits. In summary, there was a majority agreement among public and private sector R&D stakeholders in South-East Asia with the overall acceptability of TSE and its role and value in addressing the communication gap between decision makers and manufacturers. Albeit this proof of concept pilot study has shown promising early results, effort needs to be made to ensure TSE can fit within decision-making processes across LMICs and adapt to changing priorities. Moreover, considerable efforts are needed to ensure comprehensive and productive communication of the findings with R&D stakeholders and ensure a quality dialogue between the stakeholders. TSE, under the new name CAPACITI, is in the next phase of the development and issues raised in this pilot study have been critical in its further evolution,

## Supporting information

**S1 Table. Vaccine characteristics of the five hypothetical rotavirus vaccine products.**
(DOCX)

**S2 Table. Input variables required for populating the TSE model.**
(DOCX)

**S3 Table. Vaccine characteristics variables for the vaccine RVV-2 used in sensitivity analysis.**
(DOCX)

**S4 Table. Scores and ranks for the five hypothetical rotavirus vaccine products using the TSE approach.**
(DOCX)

**S5 Table. Scores and ranks for RVV-2, the second ranking vaccine product, in the base case and best case sensitivity analysis.**
(DOCX)

**S6 Table. Thematic analysis results from self-assessment forms.**
(DOCX)

**S1 File. Open-ended questionnaire distributed at initial stakeholder meeting.**
(DOCX)

**S2 File. Self-assessment form distributed at stakeholder research and development meeting.**
(DOCX)

## Acknowledgments

The authors would like to express their gratitude to Ms. Waranya Rattanavipapong, Ms. Manushi Sharma, and Mr. Md Rajibul Islam who were co-investigators in this project. Publication of study results was not contingent on the sponsor's approval or censorship of the manuscript.

## Author Contributions

**Conceptualization:** Yot Teerawattananon, Birgitte Giersing, Siobhan Botwright, Raymond C. W. Hutubessy.

**Data curation:** Rachel A. Archer.

**Formal analysis:** Rachel A. Archer, Ritika Kapoor, Yot Teerawattananon.

**Funding acquisition:** Yot Teerawattananon, Raymond C. W. Hutubessy.

**Investigation:** Rachel A. Archer, Ritika Kapoor, Yot Teerawattananon, Jos Luttjeboer.

**Methodology:** Jos Luttjeboer.

**Project administration:** Rachel A. Archer, Siobhan Botwright.

**Software:** Jos Luttjeboer.

**Supervision:** Wanrudee Isaranuwatchai, Yot Teerawattananon, Birgitte Giersing, Raymond C. W. Hutubessy.

**Validation:** Ritika Kapoor, Yot Teerawattananon.

**Visualization:** Ritika Kapoor.

**Writing – original draft:** Rachel A. Archer, Ritika Kapoor, Wanrudee Isaranuwatchai.

**Writing – review & editing:** Yot Teerawattananon, Birgitte Giersing, Siobhan Botwright, Jos Luttjeboer, Raymond C. W. Hutubessy.

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
