## [Decision Letter · Decision Letter 0]

18 Feb 2020

PONE-D-19-22507

‘It takes two to tango’: Bridging the gap between country need and vaccine product innovation

PLOS ONE

Dear Ms Archer,

Thank you for submitting your manuscript to PLOS ONE. After careful consideration, we feel that it has merit but does not fully meet PLOS ONE’s publication criteria as it currently stands. Therefore, we invite you to submit a revised version of the manuscript that addresses the points raised below during the review process.

We would appreciate receiving your revised manuscript by Apr 03 2020 11:59PM. To enhance the reproducibility of your results, we recommend that if applicable you deposit your laboratory protocols in protocols.io, where a protocol can be assigned its own identifier (DOI) such that it can be cited independently in the future. For instructions see: http://journals.plos.org/plosone/s/submission-guidelines#loc-laboratory-protocols

We look forward to receiving your revised manuscript.

Kind regards,

Ray Borrow, Ph.D., FRCPath

Academic Editor

PLOS ONE

Journal Requirements:

2. Please provide additional details regarding participant consent. In the ethics statement in the Methods and online submission information, please ensure that you have specified (1) whether consent was informed and (2) what type you obtained (for instance, written or verbal). If the need for consent was waived by the ethics committee, please include this information.

3. Please include additional information regarding the survey or questionnaire used in the study and ensure that you have provided sufficient details that others could replicate the analyses. For instance, if you developed a questionnaire as part of this study and it is not under a copyright more restrictive than CC-BY, please include a copy, in both the original language and English, as Supporting Information. Moreover, please include more details on how the questionnaire was pre-tested, and whether it was validated.

Moreover, please ensure that the models used are adequately described, and that the coding used is available.

4. "The authors would like to express their gratitude to Ms. Waranya Rattanavipapong, Ms. Manushi Sharma, and Mr. Md Rajibul Islam who were co-investigators in this project"; we would recommend that you consult our authorship requirements page: https://journals.plos.org/plosone/s/authorship, to ensure that everyone who meets our criteria for authorship  is listed as an author.

Reviewers' comments:

Reviewer's Responses to Questions

**Comments to the Author**

1. Is the manuscript technically sound, and do the data support the conclusions?

Reviewer #1: Partly

Reviewer #2: Yes

2. Has the statistical analysis been performed appropriately and rigorously? 

Reviewer #1: N/A

Reviewer #2: N/A

3. Have the authors made all data underlying the findings in their manuscript fully available?

Reviewer #1: No

Reviewer #2: Yes

4. Is the manuscript presented in an intelligible fashion and written in standard English?

Reviewer #1: Yes

Reviewer #2: Yes

5. Review Comments to the Author

Reviewer #1: The paper provides interesting insights into workshops that were organized in Thailand with stakeholders from different South East Asian countries, discussing country stakeholders’ preference for vaccine characteristics and piloting an approach called Total System Effectiveness. The authors provide a robust context and rationale for TSE and the pilot study but the methods and results descriptions are very light, making it difficult for readers to fully understand the study findings and conclusions and assess their validity. Several study tools mentioned in the paper are not described and data and data sources are not available. Some of the language used in the discussion and conclusion should be revised to avoid broad statements and better linking findings to study results. I have a list of major and minor comments below:

Major comments

1/ Introduction, lines 150-152 - The study relies on characteristics of existing vaccines and vaccines under development. The authors should comment on how the process is allowing for stakeholders to identify characteristics outside of those already suggested by the use of hypothetical products.

2/ Methods - The model used for the study should be further described, a model sketch would be useful. It’s very hard for the reader to understand what was done in the absence of a clear description or details on calculations of the model. Supplementary Table 1 provides only model variables and their description, the model input and output data (and data sources when relevant) should be made available.

3/ In L156-157 how were model criteria defined? Are these the same criteria referred to later and generated from the stakeholders workshops?

4/ L162 A description of how model outcomes were defined and how they were generated is required.

5/ L176-177 Please define the base and best case values, how they were defined and where they can be found in the article.

5/ Uncertainty in the study is not fully accounted for. A one-way sensitivity analysis is performed on vaccine characteristics of a single product. The authors should consider multivariate or probabilistic sensitivity analysis on all 5 products and characteristics or better justify their choice of a restrictive uncertainty analysis.

6/ Such as for the model, the “open ended survey” and “feedback forms” used should be described if not made available.

7/ L198 The “thematic analysis” mentioned is not described and it’s not clear from the text how it was used and if results from this analysis are shared in the paper.

8/ Results – It seems decision makers identified only criteria relevant for decision-making and not vaccine characteristics. Authors should describe how vaccine characteristics were determined, as identifying preferred vaccine characteristics is one of the objective of the study. How about providing specific vaccine attributes that were identified as ideal for a rotavirus vaccine in the Thai context?

9/ Line 209 – 210: “outcomes measures which represented the criteria are illustrated in Table 2”. Clarify if outcome measures were identified as part of the same process than criteria or if they were suggested to stakeholders. Also why results for each outcome measures and each hypothetical products are not available from the paper?

10/ L220 it is unclear what authors mean by “influencing the vaccine performance on decision criteria”. Defining how vaccine characteristics influence decision criteria is necessary to allow reader to understand the results.

11/ L222-224 If vaccine characteristics were an input to the study and not defined by it, then they should be included in the methods rather than the results.

12/ Table 3 lists decision criteria, vaccine characteristics influencing criteria and other parameters influencing criteria. I have few comments on the table:

1. Some of the items under vaccine characteristics are not really actual characteristics, for example relative risk of intussusception or method of cooling

2. The overlap between some of the vaccine characteristics across different decision criteria should be commented on. For example number of doses or vaccine schedule influencing different decision criteria.

3. The other parameters influencing criteria are broader and it’s not clear where they are coming from.

4. There should be further explanations of how or why are duration of protection or relative risk of intussusception influencing budget impact.

13/ L235-239 The results of the sensitivity analysis include characteristics that would change the ranking order of the preferred product without providing data or extent to which a characteristics would positively or negatively affect the ranking.

14/ Table 4: where are base case and best case values coming from? Does the best case refer to the best product as reported by decision makers? If yes is this a representation of the product characteristics that were shared with R&D stakeholders?

15/ L249-251 weren’t any clinical development related reasons or manufacturing considerations expressed as a trigger to particular vaccine characteristics? If not it would be interesting for the authors to comment on it.

16/ L249- 255 Not having clarity or a description of how data was collected and analyzed and the use of terms such as “stakeholders stated”, “there was broad agreement” or “it was felt that” doesn’t convey a robust scientific basis for the qualitative data results. Authors should instead consider providing results from the thematic analysis mentioned in the methods section.

17/ L267-270 The study doesn’t conclude on preferred product attributes that could inform TPP or at least they are not clearly stated, only prioritized decision criteria and broad vaccine characteristics are provided. Following the pilot experience, a reader would like to see a clear description of the ideal product, which characteristics could inform TPP meaning, how many doses, following what schedule, at what price, etc…as identified by Thai decision makers.

18/ L284 – 285 Best and worst case scenarios are not described in the methods or the results.

Minor comments

1/ Abstract L55-57 disclaimers about the use of information from anonymous stakeholders and ethics review considerations should also be reported in the text.

2/ Line 88-89 it would be interesting listing the six countries and put their relative country context in perspective with the Thai context where the pilot study was carried out.

2/ Line 116-117 “government priorities” seems broad, maybe government vaccine product preferences would be more appropriate.

3/ Line 158 change RRV5 TO RVV5

4/ Line 177 Table 4 is the first table referenced in the text

5/ L179 Three workshops were held but only two are described

6/ Table 1 is not referenced in the text and it’s not clear to which survey/form/workshops the respondents it refers to

7/ Line 208 avoid repeating “criteria” and “identified” in the same sentence

8/ Line 245 the “**” sign is not referenced in the table

9/ Line 252-253 revise the statement “preferred product characteristics of products”

10/ L273 country preferences rather than country needs

Reviewer #2: Vaccine manufacturers are extremely interested in anticipating the market demand during vaccine development. Today, evaluation of product’s desired characteristics is moving toward the earlier stages of development, and R&D activities are increasingly tailored to the target product profile (TPP) of the new vaccine. The limitation of the current approach is that, usually, vaccine manufacturers rely on the opinion of advisory groups in high income countries, which might not align with LMIC needs.

The proposed Total System Effectiveness approach (TSE), based on Multi-Criteria Decision Analysis is long overdue because it brings the tools to consider, in a systematic way, the vaccines characteristics needed by LMICs. The opinions generated by this method could be very valuable for vaccine manufacturers, provided the process is fully transparent. The argumentation of the manuscript is solid, the analytical strategy clearly supports the conclusions, and the authors actively gathered feedback from R&D stakeholders to increase the robustness of their process.

One main consideration concerns the implementation of such approach. A decision-making process should be put in place to define which studies are going to be conducted. Stakeholders from both public and private sector should be made able to influence what topics are of the greatest interest.

Suggestion is to add additional and more relevant references on the use of MCDA in vaccines such as https://doi.org/10.1016/j.vaccine.2016.10.086 and references quoted in this paper.

6. PLOS authors have the option to publish the peer review history of their article (what does this mean?). If published, this will include your full peer review and any attached files.

Reviewer #1: No

Reviewer #2: No

---

## [Author Response · Author response to Decision Letter 0]

27 Apr 2020

RESPONSES TO REVIEWERS

Reviewer's Responses to Questions

Comments to the Author

1. Is the manuscript technically sound, and do the data support the conclusions?

Reviewer #1: Partly

Reviewer #2: Yes

RESPONSE: Thank you. We have made revisions to ensure the content is technically sound.

2. Has the statistical analysis been performed appropriately and rigorously?

Reviewer #1: N/A

Reviewer #2: N/A

RESPONSE: Thank you. 

3. Have the authors made all data underlying the findings in their manuscript fully available?

Reviewer #1: No

Reviewer #2: Yes

RESPONSE: Thank you. We have revised to ensure all data is fully available. 

4. Is the manuscript presented in an intelligible fashion and written in standard English?

Reviewer #1: Yes

Reviewer #2: Yes

RESPONSE: Thank you.

5. Review Comments to the Author 

The paper provides interesting insights into workshops that were organized in Thailand with stakeholders from different South East Asian countries, discussing country stakeholders’ preference for vaccine characteristics and piloting an approach called Total System Effectiveness. The authors provide a robust context and rationale for TSE and the pilot study but the methods and results descriptions are very light, making it difficult for readers to fully understand the study findings and conclusions and assess their validity. Several study tools mentioned in the paper are not described and data and data sources are not available. Some of the language used in the discussion and conclusion should be revised to avoid broad statements and better linking findings to study results. I have a list of major and minor comments below:

Major comments

1/ Introduction, lines 150-152 - The study relies on characteristics of existing vaccines and vaccines under development. The authors should comment on how the process is allowing for stakeholders to identify characteristics outside of those already suggested by the use of hypothetical products.

RESPONSE: 

We are grateful for the reviewer's indispensable comments and thorough reading of our manuscript. The reviewer is right. In this study, we used characteristics of vaccines under development as a baseline. The study allowed for the investigators to use the baseline criteria to predict the future likelihood impact of each hypothetical vaccine product against the decision criteria identified by the stakeholders. The stakeholders directly identified the decision criteria rather than the characteristics. The researchers performed a one-way sensitivity analysis on the vaccine characteristics in order to find out the significant vaccine characteristics which could enable the vaccine product with the 2nd rank to become the most preferred vaccine product (1st rank) against the criteria. This is shown in Table 3 on the paper. The significant vaccine characteristics were found to be relative risk of intussusception, vaccine efficacy, number of doses, duration of protection, commodity cost. The best-case scenario of each of these significant parameters could independently make RVV-2 the top ranking vaccine. We have added this explanation between lines 181 and 186 (pages 7-8).

“A one-way sensitivity analysis was conducted on the vaccine characteristics, to identify the thresholds of each characteristic that could modify the overall ranking of products. It is hypothesised that this could form the basis for identifying minimum and aspirational bounds for target product profiles that meet country needs. We chose the vaccine product which is currently being ranked the second in the MCDA analysis, and via the sensitivity analysis, investigated if by varying the vaccine characteristics, it could become the top ranked vaccine.”

2/ Methods - The model used for the study should be further described, a model sketch would be useful. It’s very hard for the reader to understand what was done in the absence of a clear description or details on calculations of the model. Supplementary Table 1 provides only model variables and their description, the model input and output data (and data sources when relevant) should be made available.

RESPONSE: 

The reviewer has made a valid point. An open-access publication by Rattanavipapong et al (2020) provides a detailed description of the model in a supplementary file. The paper is open-access and therefore this description is publically available. We have referenced this paper in the text between lines 179-180 (page 7) stating that “a full description of the model and how the model outcomes were generated is available elsewhere.”

Rattanavipapong, W., Kapoor, R., Teerawattananon, Y., Luttjeboer, J., Botwright, S., Archer, R. A., ... & Hutubessy, R. C. (2020). Comparing 3 approaches for making vaccine adoption decisions in Thailand. International Journal of Health Policy and Management. Doi: 10.15171/ijhpm.2020.01

3/ In L156-157 how were model criteria defined? Are these the same criteria referred to later and generated from the stakeholders’ workshops?

RESPONSE: 

Yes the reviewer is correct. The criteria was elected by stakeholders during the first stakeholder meeting convened in May 2018 with representatives from the National Immunization Technical Advisory Group (NITAG) drug and vaccine decision-making committees, academia, pharmaceutical industry and other ministerial departments. During the workshop, stakeholders were asked to complete an open-ended survey to nominate top criteria important for vaccine product selection. Responses were collected from 15 respondents and the top five criteria were selected as the decision-making criteria for the pilot study in Thailand. We have added the following paragraph to clarify this point (lines 159-168, page 7): 

“For the purpose of the study, an excel-based model for rotavirus product selection, based on the TSE approach, was modified for the Thai context. The TSE rotavirus product selection model included a generic set of decision criteria, with defined indicators, and enabled comparison between five hypothetical rotavirus vaccine products (RVV1 to RRV5, product characteristics detailed in S1 Table). The final list of decision criteria for the Thai model was identified by stakeholders in Thailand. The final five criteria and associated outcomes for the Thai model are as follows: 1) health outcomes (i.e. cases averted, hospitalisation cases averted, and deaths averted due to the vaccination); 2) cost estimates (total programme costs, healthcare costs, and a five-year budget impact); 3) safety data (intussusception cases); 4) budget impact; and 5) cost effectiveness. Since budget impact and cost-effectiveness had not been included in the generic TSE rotavirus model, they were added for the purpose of the exercise.”

4/ L162 A description of how model outcomes were defined and how they were generated is required.

RESPONSE: 

We agree with the reviewer that this needs to be further explained. A published paper by Rattanavipapong et al. (2020) details the methodology behind the outcome estimation. This paper is open-access and therefore this description is publically available. We have referenced this paper in the text between lines to 179-180 (page 7) “a full description of the model and how the model outcomes were generated is available elsewhere.” 

5/ L176-177 Please define the base and best case values, how they were defined and where they can be found in the article.

RESPONSE: 

Following the reviewer’s suggestion, we have added supplementary file (S3 Table) which provides details of the values and the source rationale for the values for base case and the best case scenario. We have added the following sentence on lines 189-190 (page 8). “The details on the inputs for the vaccine related variables for the base case and the best case can be found in the S3 Table.” 

5/ Uncertainty in the study is not fully accounted for. A one-way sensitivity analysis is performed on vaccine characteristics of a single product. The authors should consider multivariate or probabilistic sensitivity analysis on all 5 products and characteristics or better justify their choice of a restrictive uncertainty analysis.

RESPONSE: 

The reviewer makes an interesting point here. We opted to use a one-way sensitivity analysis because it is in line with the standard approach for early health technology assessment (HTA). The approach is known as the headroom method and is a way of estimating the maximum reimbursable price of the new device over a comparator to determine a value-based price ceiling. IJzerman et al (2017) cite this as standard methodology used in early HTA. 

Whilst multivariate and probabilistic sensitivity analysis are widely accepted in traditional HTA, to our knowledge, the authors are not aware of any guidance on how these two methods can be applied to early HTA. This paper does not explore other potential sensitivity analysis approaches but this is a recommendation for future research. We have added the above justification for the use of a one-way sensitivity analysis in the discussion section on lines 334-340 (page 16). 

IJzerman, MJ, Koffijberg H, Fenwick E, Krahn M. Emerging use of early health technology assessment in medical product development: a scoping review of the literature. PharmacoEconomics. 2017;35(7):727–40

6/ Such as for the model, the “open ended survey” and “feedback forms” used should be described if not made available.

RESPONSE: 

We wholly agree with the reviewer and have made both instruments available in the supplementary files (S1 File and S2 File). 

7/ L198 The “thematic analysis” mentioned is not described and it’s not clear from the text how it was used and if results from this analysis are shared in the paper.

RESPONSE:

 This section was updated according to the reviewers’ feedback. The authors have explained the approach to thematic analysis in further detail and have added the detailed results of the thematic analysis in S6 Table. Please see lines 203-205 (page 8), “thematic analysis was deployed to analyze main patterns in the responses within the questionnaires. The forms were read repeatedly to enhance the overall understanding, then coded and classified”.

8/ Results – It seems decision makers identified only criteria relevant for decision-making and not vaccine characteristics. Authors should describe how vaccine characteristics were determined, as identifying preferred vaccine characteristics is one of the objective of the study. How about providing specific vaccine attributes that were identified as ideal for a rotavirus vaccine in the Thai context?

RESPONSE: 

The reviewer is correct. This study used the characteristics of existing or pipeline products as a baseline to predict the future likelihood impact of each vaccine against the decision criteria identified by stakeholders.

We have explained further between lines 181 and 186 (page 7-8) that:

“A one-way sensitivity analysis was conducted on the vaccine characteristics, to identify the thresholds of each characteristic that could modify the overall ranking of products. It is hypothesised that this could form the basis for identifying minimum and aspirational bounds for target product profiles that meet country needs. We chose the vaccine product which is currently being ranked the second in the MCDA analysis, and via the sensitivity analysis, investigated if by varying the vaccine characteristics, it could become the top ranked vaccine.”

We performed a one-way sensitivity analysis on vaccine characteristics to see whether this was a feasible approach to identifying the preferred product attributes for the stakeholders. We found that varying either of the following vaccines characteristic independently by giving it the best-case value (based on the most optimistic scenario) i. relative risk of intussusception, ii. number of doses iii. vaccine efficacy iv. duration of protection v. commodity cost could turn the second-ranked vaccine into first ranked in the rotavirus product selection model. This demonstrated the significance of these characteristics to Thai stakeholders in this trade-off exercise. We have added the above explanation to the discussion section on lines 288 to 295 (page 14). 

From Table 3, the most ideal or desirable vaccine product for the Thai stakeholder would have: 

• lowest relative risk of intussusception (best case = 1.0)

• highest vaccine efficacy (best case = 100%), 

• lowest number of doses (best case = 1), 

• highest duration of protection (best case = 156 weeks), 

• lowest commodity cost (best case = 1.1 USD). 

This result above can be predicted just by examining the criteria elicited by stakeholders without the modelling exercise. While this is product is most ideal it is fair to say it is probably not realistic. This was a proof-of-concept study to demonstrate whether an MCDA approach to product selection could evaluate trade-offs in characteristics for pipeline products if country stakeholder perspectives (criteria) are known.

9/ Line 209 – 210: “outcomes measures which represented the criteria are illustrated in Table 2”. Clarify if outcome measures were identified as part of the same process than criteria or if they were suggested to stakeholders. Also why results for each outcome measures and each hypothetical products are not available from the paper?

The reviewer has made a very good point. When stakeholders were asked to rank criteria in the consultation meeting, the investigators were shown a presentation on what outcome measure would be utilized for each criteria in the generic TSE model (see the answer to major comment /3). The outcome measures for each criteria are illustrated in Table 1. Whilst we did not ask stakeholders to rank outcome measures, implicitly the outcome measures were identified by stakeholders. The outcome measures for each criterion corresponded to indicators in the TSE rotavirus comparison model. Please consult Table 3 in Rattanavipapong et al (2020) which contains the results for each outcome measure. The outcome measures used to evaluate the vaccine performance in each criterion could be adapted to different country settings and has been added in the discussion section. However, the model to evaluate the outcome measures would need to be adapted accordingly. We have added this as a limitation in the discussion section on lines 314-317 (page 15). The paper now states “Furthermore, the outcomes associated with each criterion were already pre-determined in the model. Allowing stakeholders to define their own outcomes may lead to a better understanding of expectations for pipeline products from a country perspective.”

10/ L220 it is unclear what authors mean by “influencing the vaccine performance on decision criteria”. Defining how vaccine characteristics influence decision criteria is necessary to allow reader to understand the results.

RESPONSE:

Thanks you to the reviewer for this insightful comment. Vaccine characteristics for the five hypothetical rotavirus vaccine products (RVV1 to RRV5) including vaccine efficacy, duration of protection, dosing schedule, safety, cost, components of vaccine constitution, volume, application type, and delivery and storage requirements are noted in S1 Table. Multiple vaccine characteristics were identified which could impact the scores of the vaccine products on the decision criteria nominated by stakeholders. One vaccine characteristic could influence the vaccine performance in one or more decision criteria. Other system or country level factors which may influence overall success of the vaccine program were also recorded. For example, the decision criteria for safety was found to be influenced by the relative risk of intussusception, vaccine efficacy, number of doses and dosing schedule. Health impact was influenced by vaccine schedule, dosage, efficacy, duration of protection and also the coverage and socio-economic status of the population expected from the national program. We have provided further detail on lines 230 and 238.

A detailed description of the calculations of model outcomes and the influence of vaccine characteristics on the different decision criteria could be found in open-access publication by Rattanavipapong et al (2020) which describes the model in a supplementary file. The paper is open-access and therefore this description is publically available. We have referenced this paper in the text between lines 179-180 (page 7) stating that “a full description of the model and how the model outcomes were generated is available elsewhere.”

Rattanavipapong, W., Kapoor, R., Teerawattananon, Y., Luttjeboer, J., Botwright, S., Archer, R. A., ... & Hutubessy, R. C. (2020). Comparing 3 approaches for making vaccine adoption decisions in Thailand. International Journal of Health Policy and Management. Doi: 10.15171/ijhpm.2020.01

11/ L222-224 If vaccine characteristics were an input to the study and not defined by it, then they should be included in the methods rather than the results.

RESPONSE: 

Thank you for the reviewer for the suggestion. You are right in that the different features (or characteristics) specific to each of the hypothetical vaccine products in S1 Table were inputs in the study as mentioned in the methods. By running the model, we could identify significant characteristics which could influence the scores of the vaccine products against criteria elicited by stakeholders in this Thai case scenario. Table 3 provides a list of these influential vaccine characteristics and therefore have been reported as a result for the Thai case example. The sensitivity analysis was then conducted on the vaccine characteristics that we found to be able to impact the scores of vaccine products against the multi decision criteria. We would therefore like to keep in the results section.

12/ Table 3 lists decision criteria, vaccine characteristics influencing criteria and other parameters influencing criteria. I have few comments on the table:

1. Some of the items under vaccine characteristics are not really actual characteristics, for example relative risk of intussusception or method of cooling

RESPONSE: 

Thank you for the reviewer for this comment. We have referred all features specific to the vaccine products as vaccine characteristics. These features are expected to impact the overall cost and health benefits which can be achieved by using that vaccine product in the immunization campaign. Though some of them may not seem as the commonly referred vaccine characteristics but we still feel that they are related directly to the vaccine product and can be referred as vaccine (product) characteristics.

2. The overlap between some of the vaccine characteristics across different decision criteria should be commented on. For example number of doses or vaccine schedule influencing different decision criteria.

RESPONSE: 

The reviewer is correct, some vaccine characteristics were found to impact more than one decision criteria (lines 230-238, page 9-10). This is because the different decision criteria are not mutually exclusive and have some interdependence. Hence, varying one vaccine characteristics may impact the vaccine product ranking on multiple decision criteria. For example, the number of doses of vaccine is found to have an influence on 4 decision criteria- safety, health impact, budget impact and cost-effectiveness. This is because, vaccine which has higher number of doses are expected to not provide complete protection after the first dose and is also expected to have a lower adherence as individuals are required to go to the healthcare institution for multiple doses. A lower initial protection/ adherence to the vaccine doses leads to a suboptimal coverage among the population and is expected to lead to an increase in rotavirus cases in comparison to a comparator with a single dose which imparts full protection in the first dose itself. These increased cases have a direct influence on the health impact achieved, and the overall resource use for the treatment of these cases, which in turn influence the budget impact and the cost-effectiveness analysis. Also, as the risk of side effects of intussusception are highest post vaccine administration, a vaccine with multiple doses may have an overall higher risk of intussusception, which would influence its performance on the criterion of safety.

3. The other parameters influencing criteria are broader and it’s not clear where they are coming from.

RESPONSE: 

The model uses some local parameters along with the vaccine characteristics to generate outcomes for the different vaccine products (see detailed explanation on lines 159-174, page 7). These locally relevant parameter inputs have been generated from government reports, published literature and expert opinions. They include relevant parameters included local inputs on socio-economic status, coverage of vaccine programs, vaccine efficacy, and schedule, costs for storage, training and administration, and other epidemiological data on birth cohort, disease burden and resource use (lines 171-174). More details on this can be obtained from the S2 Table and in another related paper which has been published by Rattanavipapong et al (2020). 

4. There should be further explanations of how or why are duration of protection or relative risk of intussusception influencing budget impact.

RESPONSE: 

The overall budget impact of the vaccine includes the cost of the immunization program and the also the healthcare resources spent on the rotavirus cases and the intussusception cases the population experiences. Hence, if a vaccine product has a lower relative risk of intussusception or a longer duration of protection from the disease, this is expected to reduce the number of cases and hence lead to healthcare savings from the treatment of the diseases. Thus, a vaccine which has the same profile and cost but a higher duration of protection/or a lower intussusception risk from its comparators, is expected to have a relatively lower budget impact. In Table 1 in the paper, we have stated that the 5-year budget impact includes the cost of program and healthcare cost and added a brief footnote explaining this in the table.

13/ L235-239: The results of the sensitivity analysis include characteristics that would change the ranking order of the preferred product without providing data or extent to which a characteristics would positively or negatively affect the ranking.

RESPONSE: 

The reviewer has made an excellent point regarding the results of the sensitivity analysis which need to be included in the manuscript. We have added Supplementary table 5 (S5 Table) which provides details on how the scoring of the vaccine product RVV-2 varied with the different best-case scenarios used in one-way sensitivity analysis and its impact on the overall ranking. Also, text has been added in the manuscript to link the reader to the required table in the supplementary (lines 250-251, page 12). Further details on the methodology of the TSE approach can be found in the reference Rattanavipapong et al (2020).

14/ Table 4: where are base case and best case values coming from? Does the best case refer to the best product as reported by decision makers? If yes is this a representation of the product characteristics that were shared with R&D stakeholders?

RESPONSE: 

We are grateful for the reviewer's constructive comments regarding the best and base case. As previously stated we have added in an additional supplementary table (S3 Table) which explains the base case and best case in further details and the source/ rationale for the values for base case and the best case scenario. We refer to this between lines 189 and 190.

15/ L249-251 weren’t any clinical development related reasons or manufacturing considerations expressed as a trigger to particular vaccine characteristics? If not it would be interesting for the authors to comment on it.

RESPONSE: 

 The focus for the study was identifying criteria that would be used for decisions on whether to introduce a vaccine (i.e. the criteria leading to a policy decision for the health programme). As such, our study did not consider likelihood of clinical success or manufacturing feasibility. We have modified the following wording to the discussion to highlight this limitation on lines 299-302 (page 15): “As such, the TSE approach could allow manufacturers to understand country needs, which could be combined with technical considerations such as likelihood of clinical success and manufacturing feasibility to develop more robust TPPs and roadmaps for products with higher chance of country uptake that will ultimately lead to greater public health impact [19]”

Improving communication and transparency about government’s priorities from a vaccination program to the manufacturers could help them develop products with higher chances of inclusion in clinical practice. This study focuses on identifying the influential vaccine characteristics based on country priorities, and helping the R&D identify the target product portfolio for their products. As different countries may have different challenges to overcome like lack of trained staff, storage constraints, non-compliance among population etc., it is very important for manufacturers to understand the country needs better while investing huge amount of resources in product development. However, post this the manufacturers would need to perform an internal evaluation and identify the challenges with clinical development, technological expertise and other manufacturing considerations, which may prevent them from developing a product as per the TPP identified earlier. Understanding the country priorities forms the first step for manufacturers and is the focus of our study, and we do not discuss about the challenges experienced by the manufacturer.

16/ L249- 255 Not having clarity or a description of how data was collected and analyzed and the use of terms such as “stakeholders stated”, “there was broad agreement” or “it was felt that” doesn’t convey a robust scientific basis for the qualitative data results. Authors should instead consider providing results from the thematic analysis mentioned in the methods section.

RESPONSE: 

We thank the reviewer again for this helpful suggestion. As mentioned previously, have described how the self-assessments were thematically analyzed in further detail on lines 203 and 205 and have also added a supplementary file that summarizes the results of the thematic analysis in S6 Table. We direct the reader to this supplementary file in the results section. 

17/ L267-270 The study doesn’t conclude on preferred product attributes that could inform TPP or at least they are not clearly stated, only prioritized decision criteria and broad vaccine characteristics are provided. Following the pilot experience, a reader would like to see a clear description of the ideal product, which characteristics could inform TPP meaning, how many doses, following what schedule, at what price, etc…as identified by Thai decision makers.

RESPONSE: 

The authorship team are not quite clear if the reviewer is referring to the ideal product for rotavirus vaccine or all vaccine products. If the latter, then we do not feel this is possible as the preferred product characteristics of a vaccine depends on the type of vaccine and we are against the notion that one size fits all. 

Regarding the ideal characteristics of rotavirus vaccine for Thai stakeholders, this is a very important point made by the reviewer, we hope that we have answered this query to you satisfaction in the response to Comment 8/ Results.

18/ L284 – 285 Best and worst case scenarios are not described in the methods or the results.

RESPONSE: 

Many thanks. As stated above, we have added a supplementary table (S3 Table) and text in the manuscript lines 189 and 190 (page 8) that describes the best and base case scenarios in further detail. 

Minor comments

1/ Abstract L55-57 disclaimers about the use of information from anonymous stakeholders and ethics review considerations should also be reported in the text.

RESPONSE: 

Thank you for the suggestion, we have these points in relation to ethics to the text on lines 201-203 (page 8).

2/ Line 88-89 it would be interesting listing the six countries and put their relative country context in perspective with the Thai context where the pilot study was carried out.

RESPONSE: 

This paper is focusing primarily on the case of Thailand, so while this is an interesting angle we do not feel like it will strengthen the paper. Moreover, we may not be able to showcase the nuances of these country contexts. The authors believe it would be more useful for readers to refer to the paper cited. We are however appreciative of the reviewers’ suggestion. 

2/ Line 116-117 “government priorities” seems broad, maybe government vaccine product preferences would be more appropriate.

RESPONSE: 

Thank you. The authorship team have changed the wording as suggested.

3/ Line 158 change RRV5 TO RVV5

RESPONSE: 

This correction has been made.

4/ Line 177 Table 4 is the first table referenced in the text

RESPONSE: 

Thank you to the reviewer for their detailed review. We have now changed the order of the tables. 

5/ L179 Three workshops were held but only two are described

RESPONSE: 

Thank you to the reviewer for noticing this. A third meeting was convened to present the results of the pilot studies back to the stakeholders who attended the initial meeting in May. We have added a sentence about the third meeting on lines 206 to 207 (page 8). 

6/ Table 1 is not referenced in the text and it’s not clear to which survey/form/workshops the respondents it refers to

RESPONSE: 

Thank you. We have now referenced the table in the text and added the following title to the table “Profile of respondents from self-assessment forms”. This is now Table 4.

7/ Line 208 avoid repeating “criteria” and “identified” in the same sentence

RESPONSE: 

We wholly agree with the reviewer that this sentence does not flow well as it stands. We have revised the sentence to read as follows “the top 5 decision-making criteria identified by stakeholders for choosing between rotavirus vaccine products were as follows: health impact, safety, budget impact, cost-effectiveness and delivery cost.” (Lines 213-214, pages 8-9)

8/ Line 245 the “**” sign is not referenced in the table

RESPONSE: 

Thank you. We have revised accordingly. 

9/ Line 252-253 revise the statement “preferred product characteristics of products”

RESPONSE: 

The authorship have corrected this following the reviewers’ suggestion.

10/ L273 country preferences rather than country needs

RESPONSE: 

We changed this as advised.

REVIEWER #2:

Vaccine manufacturers are extremely interested in anticipating the market demand during vaccine development. Today, evaluation of product’s desired characteristics is moving toward the earlier stages of development, and R&D activities are increasingly tailored to the target product profile (TPP) of the new vaccine. The limitation of the current approach is that, usually, vaccine manufacturers rely on the opinion of advisory groups in high income countries, which might not align with LMIC needs.

The proposed Total System Effectiveness approach (TSE), based on Multi-Criteria Decision Analysis is long overdue because it brings the tools to consider, in a systematic way, the vaccines characteristics needed by LMICs. The opinions generated by this method could be very valuable for vaccine manufacturers, provided the process is fully transparent. The argumentation of the manuscript is solid, the analytical strategy clearly supports the conclusions, and the authors actively gathered feedback from R&D stakeholders to increase the robustness of their process.

One main consideration concerns the implementation of such approach. A decision-making process should be put in place to define which studies are going to be conducted. Stakeholders from both public and private sector should be made able to influence what topics are of the greatest interest.

Suggestion is to add additional and more relevant references on the use of MCDA in vaccines such as https://doi.org/10.1016/j.vaccine.2016.10.086 and references quoted in this paper.

RESPONSE:

We thank the reviewer for their invaluable feedback and optimistic stance. We have added the reference Knobler et al (2017) in addition to Madhavan et al (2015) to the discussion section between lines 305 and 308 (page 15).

---

## [Decision Letter · Decision Letter 1]

18 May 2020

‘It takes two to tango’: Bridging the gap between country need and vaccine product innovation

PONE-D-19-22507R1

Dear Dr. Archer,

We are pleased to inform you that your manuscript has been judged scientifically suitable for publication and will be formally accepted for publication once it complies with all outstanding technical requirements.

With kind regards,

Ray Borrow, Ph.D., FRCPath

Academic Editor

PLOS ONE

Additional Editor Comments (optional):

Reviewers' comments:

Reviewer's Responses to Questions

**Comments to the Author**

1. If the authors have adequately addressed your comments raised in a previous round of review and you feel that this manuscript is now acceptable for publication, you may indicate that here to bypass the “Comments to the Author” section, enter your conflict of interest statement in the “Confidential to Editor” section, and submit your "Accept" recommendation.

Reviewer #1: All comments have been addressed

Reviewer #2: All comments have been addressed

2. Is the manuscript technically sound, and do the data support the conclusions?

Reviewer #1: Yes

Reviewer #2: (No Response)

3. Has the statistical analysis been performed appropriately and rigorously? 

Reviewer #1: Yes

Reviewer #2: (No Response)

4. Have the authors made all data underlying the findings in their manuscript fully available?

Reviewer #1: Yes

Reviewer #2: Yes

5. Is the manuscript presented in an intelligible fashion and written in standard English?

Reviewer #1: Yes

Reviewer #2: Yes

6. Review Comments to the Author

Reviewer #1: (No Response)

Reviewer #2: (No Response)

7. PLOS authors have the option to publish the peer review history of their article (what does this mean?). If published, this will include your full peer review and any attached files.

Reviewer #1: No

Reviewer #2: No

---

## [Editor Report · Acceptance letter]

1 Jun 2020

PONE-D-19-22507R1 

‘It takes two to tango’: Bridging the gap between country need and vaccine product innovation 

Dear Dr. Archer:

I am pleased to inform you that your manuscript has been deemed suitable for publication in PLOS ONE. Congratulations! Your manuscript is now with our production department. 

With kind regards,

on behalf of

Prof. Ray Borrow 

Academic Editor

PLOS ONE